# Locally commensurate charge-density wave with three-unit-cell periodicity in YBa$_2$Cu$_3$O$_y$

Igor Vinograd [1]✉, Rui Zhou [1,4], Michihiro Hirata[1,5], Tao Wu[1,6], Hadrien Mayaffre[1], Steffen Krämer [1], Ruixing Liang[2,3], W. N. Hardy[2,3], D. A. Bonn[2,3] & Marc-Henri Julien [1]✉

In order to identify the mechanism responsible for the formation of charge-density waves (CDW) in cuprate superconductors, it is important to understand which aspects of the CDW's microscopic structure are generic and which are material-dependent. Here, we show that, at the local scale probed by NMR, long-range CDW order in YBa$_2$Cu$_3$O$_y$ is unidirectional with a commensurate period of three unit cells ($\lambda = 3b$), implying that the incommensurability found in X-ray scattering is ensured by phase slips (discommensurations). Furthermore, NMR spectra reveal a predominant oxygen character of the CDW with an out-of-phase relationship between certain lattice sites but no specific signature of a secondary CDW with $\lambda = 6b$ associated with a putative pair-density wave. These results shed light on universal aspects of the cuprate CDW. In particular, its spatial profile appears to generically result from the interplay between an incommensurate tendency at long length scales, possibly related to properties of the Fermi surface, and local commensuration effects, due to electron-electron interactions or lock-in to the lattice.

[1] Univ. Grenoble Alpes, INSA Toulouse, Univ. Toulouse Paul Sabatier, EMFL, CNRS, LNCMI, Grenoble, France. [2] Department of Physics and Astronomy, University of British Columbia, Vancouver, BC, Canada. [3] Canadian Institute for Advanced Research, Toronto, Canada. [4] Present address: Institute of Physics, Chinese Academy of Sciences, and Beijing National Laboratory for Condensed Matter Physics, Beijing, China. [5] Present address: MPA-Q, Los Alamos National Laboratory, Los Alamos, NM, USA. [6] Present address: Hefei National Laboratory for Physical Sciences at the Microscale, University of Science and Technology of China, Hefei, Anhui, China. ✉email: grvngrd@gmail.com; marc-henri.julien@lncmi.cnrs.fr

                                   1

In recent years, charge order, a periodic modulation of the charge density and lattice positions, called charge-density wave (CDW), has been shown to be a universal property of hole- and electron-doped cuprates[1], with possible connections to the pseudogap and superconducting phases[2–9]. However, several fundamental questions remain unanswered such as why there are different CDW phases (with or without long-range order, with or without intertwined magnetic order) or what sets the CDW wave vector (and the associated periodicity in real space).

More generally, the knowledge of the spatial profile of the charge modulation, which includes its periodicity but also its intra-unit-cell structure and the possible presence of topological defects or discommensurations, is important for understanding quantum oscillations[10,11], the coexistence of the CDW with superconductivity[12–15] or the coupling between spin and charge degrees of freedom. For instance, the latter may depend on whether the CDW periodicity is odd or even, given that $CuO_2$ planes constitute a bipartite lattice of antiferromagnetic moments.

According to scattering experiments, Y-, Bi- and Hg-based cuprates show short-range, bi-directional (albeit likely unidirectional at short length scales[16–18]) CDW order with an incommensurate wave vector $q$ that decreases upon increasing hole doping $p$[1]. There is no associated spin-order (the case of cuprates with intertwined spin and charge orders will be alluded to in the discussion section). Qualitatively, both the $p$ dependence and the incommensurate nature of $q$ are naturally understood if the ordering wave vector connects parts of the Fermi surface, such as the parallel segments near the antinodes (such nesting favours a CDW instability in some models, e.g.[19]) or the hot spots, those special points where the Fermi surface intersects the anti-ferromagnetic Brillouin zone (in which case the origin of the CDW lies in antiferromagnetic correlations[20,21]). Quantitatively, however, whether the experimentally measured CDW wave vector relates to the Fermi surface geometry is under debate[3,22].

This view has been challenged by recent progress in the analysis of scanning tunnelling microscopy (STM) measurements in Bi-based cuprates revealing an extended doping range in which the local wave vector $q_0$ is 1/4 (in units of $\frac{2\pi}{a}$, with $a$ being the crystal lattice parameter), equivalent to a local commensurate period $\lambda = 4a$[23–27]. Mesaros et al. have taken this as evidence that there is a universal instability towards the formation of a period-four density wave, fundamentally rooted in strong-correlation physics[23].

Among cuprates showing incommensurate short-range CDW order and no spin-order, $YBa_2Cu_3O_y$ (YBCO) is distinguished by (1) larger values of the CDW correlation length in zero-field and (2) the presence of three-dimensional (3D) long-range CDW order, observed by quenching superconductivity in high magnetic fields[4,28–35] or applying strain to the sample[18,36,37]. According to X-ray scattering studies, the 3D CDW is unidirectional (propagating along the $b$ axis, parallel to the chain direction) with the same incommensurate wave vector $q_b \simeq 0.3$ as the short-range CDW in zero field[31–33,37].

Here, we present supporting evidence from $^{17}O$ NMR that the 3D CDW phase of YBCO has a commensurate local period of three unit cells, not four as in Bi-based cuprates, over an extended range of hole doping $p$ and magnetic field values, $B_z$, where $\hat{z}$ is the unit vector perpendicular to the $CuO_2$ planes. We discuss how discommensurations (phase slips) reconcile this observation with the scattering results. Taken together with the STM data in Bi-based cuprates, our measurements indicate that the spatial profile of the cuprate CDW generically results from the interplay between an incommensurate tendency at long length scales, and local commensuration effects due to electron–electron interactions, as proposed in ref.[23], or lock-in to the lattice. We also show how $^{63}Cu$ NMR data, initially interpreted as evidence of period-4 CDW (ref.[28] and Supplementary Note 1), can be reinterpreted as a period-3 CDW, though this entails that the CDW amplitude must be greatly reduced below empty chains.

## Results

**CDW characteristics encoded in NMR spectra.** Like STM, NMR is a local probe and is sensitive to the local periodicity and other aspects of the modulation. Specifically, the NMR line shape represents the combined histogram of both the local magnetic hyperfine fields, characterised by the Knight shift $K$, and the local electric field gradients (EFG), encoded in the effective quadrupole frequency $\nu_{quad}(\mathbf{r})$ at positions $\mathbf{r}$ in the bulk of a sample. Note that $\nu_{quad}$ is defined here as half the distance between first low and high-frequency satellite lines, hereafter called LF1 and HF1, in any given field orientation. Since both $K$ and $\nu_{quad}$ may depend on the local charge density $\rho(\mathbf{r})$, the line shape potentially contains rich information on the CDW[38]. In order to obtain this information, however, one first needs to understand to what extent the NMR spectrum represents a faithful histogram of $\rho(\mathbf{r})$ values. This is not necessarily so in the following circumstances:

First, the periodic lattice distortion associated with the CDW may also affect both $K(\mathbf{r})$ and $\nu_{quad}(\mathbf{r})$. Also, a change in the symmetry of the local charge distribution may affect the symmetry of the EFG tensor and thus $\nu_{quad}$. Here, we neglect these effects and instead assume that the CDW-induced changes $\Delta K(\mathbf{r})$ and $\Delta \nu_{quad}(\mathbf{r})$ essentially reflect the amplitude of the charge density modulation at the site of the probed nucleus. This is reasonable given the large contribution of the on-site hole density to the EFG at both Cu and O sites in the $CuO_2$ planes[39,40].

Second, $\Delta K(\mathbf{r})$ and $\Delta \nu_{quad}(\mathbf{r})$ may, in principle, not be linear functions of $\rho(\mathbf{r})$. We shall examine this possibility below and conclude that it is irrelevant here.

Third, depending on the relative magnitude of the two effects and on which line is considered, the changes in $K(\mathbf{r})$ and $\nu_{quad}(\mathbf{r})$ may, partially or entirely, compensate each other. In YBCO, there are five different lines per $^{17}O$ site (nuclear spin $I = 5/2$) and three per $^{63}Cu$ (nuclear spin $I = 3/2$) and our previous work[28,41] has shown how each line shape could be quantitatively understood from the combination of magnetic and quadrupole effects. We have established that the effects of the CDW are better seen on the upper frequency quadrupole satellites of $^{63}Cu$ and $^{17}O$, hereafter called HF1 and HF2 respectively (see Methods for details).

Therefore, we consider that the high-frequency satellite transitions provide a direct image of the charge-density modulation in the $CuO_2$ planes of YBCO, which will be corroborated by the overall consistency of our $^{17}O$ NMR results with both the $^{63}Cu$ results and the information from X-ray scattering.

Suppose a unidirectional modulation of the charge density $\rho$ at atomic positions $x$, having an amplitude $\Delta \rho$ around an average charge density $\rho_0$, a phase $\phi$ and propagating with a wave vector $q$, parallel to the $b$ axis in our case:

$$\rho(x) = \rho_0 + \Delta \rho \cos(q \cdot x + \phi). \qquad (1)$$

Figure 1 shows that the histograms for incommensurate and commensurate unidirectional modulations are quite different, even though their periods are close ($\lambda = 3.436b$ and $3.0b$, corresponding to $|q| = 2\pi/\lambda = 0.291$ and 1/3, respectively). More generally, NMR line shapes are very sensitive to all microscopic aspects of the CDW: the modulation amplitude $\Delta \rho$ and the period but also the phase $\phi$, as exemplified in Supplementary Note 2 and Supplementary Fig. 2. Any deviation from a sinusoid will further affect the spectrum but non-sinusoidal (e.g. solitonic) forms will not be considered here. Furthermore, as the comparison of Fig. 1 and Supplementary Fig. 3 shows the NMR line shapes are in most

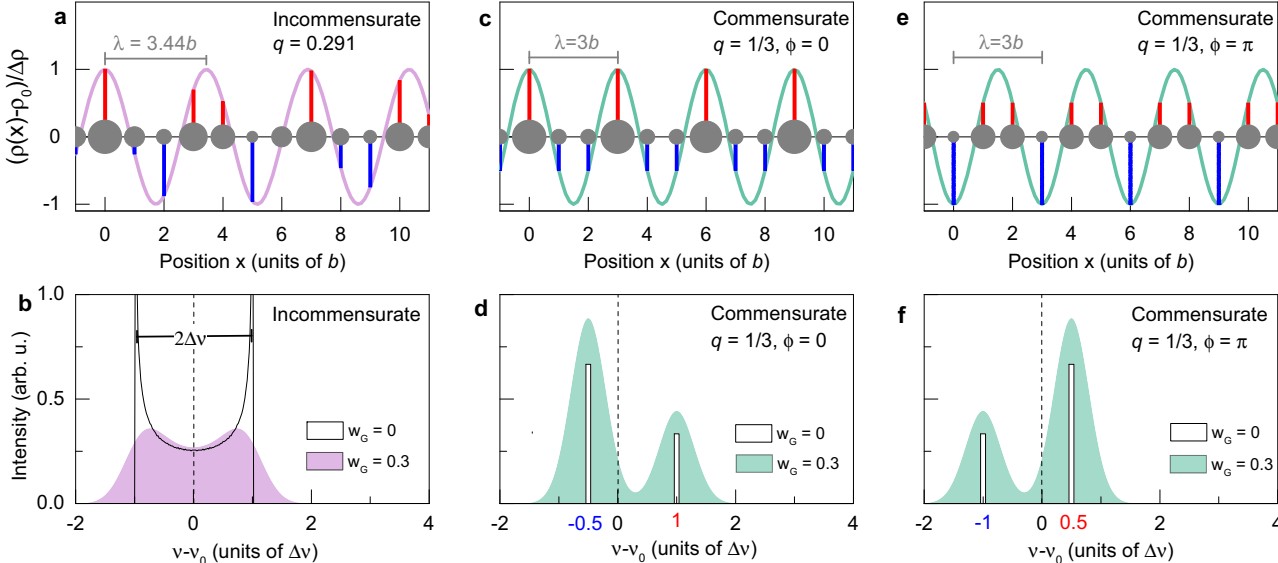

**Fig. 1 NMR line shapes for incommensurate and commensurate unidirectional CDWs. a** Incommensurate sinusoidal charge modulation $\rho(x) = \rho_0 + \Delta\rho \cos(q \cdot x + \phi)$ with period $\lambda = 2\pi/|q| = 3.436b$. Grey dots of different sizes depict atoms with varying charge density. Red and blue bars mark the sampled positive and negative CDW amplitude at the atomic positions (separated by the lattice constant $b$). **b** Black line: characteristic histogram for an incommensurate CDW, determined by sampling the modulation shown in (**a**) at the atomic positions: there are two singularities (equally shifted from the 0 position that corresponds to the resonance position $\nu_0$ in the absence of a CDW) and a continuum of intensity in between the singularities since the incommensurate nature implies that the charge modulation is sampled by nuclei at an infinite number of different values in between the two extrema, regardless of the values of both the period and the phase. Any incommensurate wave vector, whatever its value, leads to the exact same NMR line shape. For this figure, we thus use $q = 0.291$, a value slightly smaller than the experimental value $q = 0.323$ for $p = 0.11$, because this makes it easier to visualise the inequivalent sites in the 11-site chain segment (see also Supplementary Fig. 1). The frequency $\nu(x)$ is proportional to the CDW amplitude $\rho(x) - \rho_0$. Convolution with a Gaussian of width $w_G = 0.3$ (in units of $\Delta\nu$ where $\Delta\nu$ is the frequency shift for a site at the maximum of the modulation) gives the coloured symmetric line shape, resembling the sum of two peaks of equal amplitude. **c** Commensurate sinusoidal modulation for $\lambda = 3b$ and $\phi = 0°$. **d** Histogram of the CDW amplitudes for the commensurate modulation in (**c**) with Gaussian broadening ($w_G = 0.3$) and without ($w_G = 0$): there are two discrete peaks with an area ratio of 2:1 and shift ratio $-1$:2. **e** Same modulation as in (**c**) but with a $\pi$ phase shift (equivalent to a translation by $3b/2$). **f** Histogram corresponding to the modulation sampled in (**e**) with and without Gaussian broadening ($w_G$).

cases very different for unidirectional and bi-directional modulations (even though the overall spread in frequency remains a measure of $\Delta\rho$ in all cases).

Although it lacks the atomic-scale imaging capability of STM, NMR is also able to provide atomically resolved information by probing different nuclei and/or different crystallographic sites, and it does so in the bulk, not on the surface, over broad ranges of temperature and field. This allows, for instance, to single out distinct CDWs in different planes of a layered material[42]. Here in YBCO, we shall exploit our data at O and Cu sites to discuss whether the CDW phase $\phi$ varies between different atomic positions within the CuO$_2$ unit cell. From the purely experimental point of view, $^{17}$O NMR has a number of advantages with respect to $^{63}$Cu NMR: better signal-to-noise ratio, more reliable signal intensity (see Methods) and, as we shall show, a greater sensitivity to the CDW.

**Site-selective aspects in YBCO.** We primarily (but not exclusively) focus on data from YBa$_2$Cu$_3$O$_{6.56}$. At this hole doping level ($p = 0.109$), oxygen dopants form chains (parallel to the $b$ axis) every other site along $a$. This (so-called ortho-II or O-II) phase is the simplest and longest-range form of oxygen order in the phase diagram. This leads to the narrowest NMR lines and the simplest NMR spectra, with only three inequivalent O sites in the CuO$_2$ planes, as depicted in Fig. 2a: a single O(2) site on Cu-O-Cu bonds along the crystallographic $a$ axis and two inequivalent O(3) sites, called O(3F) and O(3E) (below full and empty chains, respectively), along the $b$ axis[29,41,43].

As O(3F) and O(3E) lines overlap in the current experimental conditions (low temperature and $B$ nearly $\parallel c$, see Supplementary

Fig. 4), we focus on O(2) data that are much less difficult to analyse.

**Commensurate local period from $^{17}$O NMR.** The experimental O(2) NMR line shape for $p = 0.109$ splits into a more intense peak (O$_A$) and a less intense peak (O$_B$) (Fig. 3), meaning that the CDW creates only two inequivalent O(2) sites.

Such a line shape is in general inconsistent with bi-directional modulations, as shown in Supplementary Note 3 and Supplementary Fig. 3. This is no surprise, as the unidirectional character of the high-field CDW was already established from the observation of similarly split $^{63}$Cu line shapes[28] as well as from X-ray scattering[31–35]. Experimentally, the total charge modulation is not purely unidirectional in high fields since the long-range unidirectional CDW coexists with the short-range bi-directional CDW[17,31]. However, this latter only broadens the NMR spectra symmetrically[17] and so does not modify the overall line shape. From now on, we thus only discuss unidirectional modulations.

X-ray scattering in high fields finds an incommensurate wave vector $q = 0.323 \pm 0.005$ for YBCO with $p \simeq 0.11$. This should give a spectrum with two peaks of equal intensities and an asymmetric shape of the individual peaks, as shown in Fig. 1b. Notice that we use $q = 0.291$, a value slightly smaller than the experimental value $q = 0.323$, because this makes it easier to visualise the inequivalent sites in the 11-site chain segment in Fig. 1a. This has no impact on the simulated line shape in Fig. 1b since the histogram is independent of the value of $q$, provided it is incommensurate. The line shape only depends on the

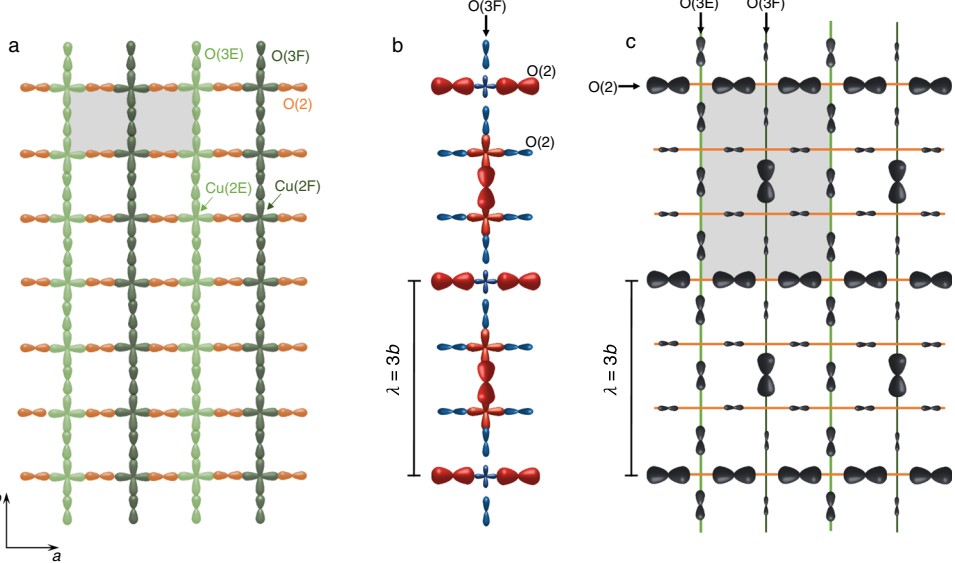

**Fig. 2 Crystallographic sites and possible CDW pattern in CuO$_2$ planes of ortho-II YBCO. a** Crystallographic structure with inequivalent sites in CuO$_2$ planes with alternating rows of Cu(2E) and O(3E) below empty chains (light green) and rows of Cu(2F) and O(3F) below oxygen-full chains (dark green). The grey rectangle shows the crystallographic unit-cell in the O-II structure. **b** Charge distribution pattern below full chains compatible with O(2) and Cu NMR line shapes. The size of the orbitals approximately scales with the charge density while the red (blue) colours represent positive (negative) variations with respect to the average density (same red/blue colour code as in Figs. 1, 3). The charge pattern at Cu sites has a phase shift $\delta\phi = \pi$ with respect to O(2) sites, consistent with our analysis of the line shapes in Fig. 3 (see text). The modulation at O(3F) is chosen to be in-phase with Cu(2F), implying a $d$ form factor ($\delta\phi = \pi$) between O(3F) and O(2) (see discussion in text). **c** Larger field of view of the charge density pattern in a CuO$_2$ plane, showing only the O sites (Cu sites are omitted). The grey rectangle shows the electronic unit cell. The depicted pattern is a simplified version of the actual modulation in two respects: (1) the modulation is depicted with the same amplitude for O(3F) and O(2) sites while the amplitude is likely to be different (presumably smaller at O(3) sites by a factor up to 3, see Supplementary Note 4). (2) The modulation amplitude at O(3E) and Cu(2E) sites, i.e. below empty chains, is set to zero but it may only be smaller than below full chains (by a factor of at least 3) as both possibilities are compatible with the absence of visible splitting of the Cu(2E) line.

dimensionality (whether the CDW is uni- or bi-directional) and on the shape of the modulation (sinusoidal, square-like, solitonic-type, etc.). Clearly, the $^{17}$O-NMR line shape is not compatible with an incommensurate modulation if one assumes that the NMR Knight shift and the quadrupole shift are linearly coupled to the charge density. In Supplementary Note 6 and Supplementary Fig. 7, we also show that including a quadratic term in the coupling, while indeed producing peaks of unequal intensities[38], does not describe the data adequately. The unequal peak intensities are also inconsistent with $\lambda = 4b$ and more generally with any even period (Supplementary Fig. 2).

On the other hand, a unidirectional commensurate modulation with $\lambda = 3b$ and $\phi = 0°$ (Fig. 1c) leads to a histogram that has two discrete peaks with an area ratio of 2:1 (Fig. 1d). The peak with smaller area (B) is shifted by $+\Delta\nu$ while the larger peak (A) is shifted by $-0.5\Delta\nu$ in the opposite direction (i.e. the A:B shift ratio is $-1$:2), with respect to the resonance position in the absence of CDW. While this histogram visually resembles the experimental spectrum (Fig. 3), precise fitting is required for a quantitative comparison.

The procedure, detailed in "Methods", Supplementary Note 5 and Supplementary Figs. 5, 6, consists of simultaneous fitting of the four quadrupole satellites (i.e. not only the HF2 spectrum shown in Fig. 3), each with two asymmetric line shapes of equal widths. The asymmetry of the individual $^{17}$O peaks, which is essential for a reliable fit, has been shown to arise from an inhomogeneous pattern of the local density of states (LDOS) at the Fermi level that develops alongside the high-field CDW[41] (a recent theoretical study further relates this inhomogeneous LDOS to the curvature of the Fermi surface at the hot spots[44]). The asymmetry is determined from peaks that do not split and is thus

not a fit parameter[41]. The fit parameters are the area and the frequency of each peak. This provides us with two useful parameters to benchmark the model: the relative area of the two sets of peaks and their quadrupole shift ratio:

$$\frac{\Delta\nu_{quad,B}}{\Delta\nu_{quad,A}} = \frac{\nu_{quad,B} - \nu_{quad,0}}{\nu_{quad,A} - \nu_{quad,0}} \qquad (2)$$

where $\nu_{quad,A}$ and $\nu_{quad,B}$ are the quadrupole frequencies defined by the peak positions and $\nu_{quad,0}$ is the quadrupole frequency outside of the high-field phase (at low field and/or high temperature).

Fits to the data at $B_z = 27.1$ T (Fig. 3a and Supplementary Fig. 6) indicate that the area ratio of the peaks is $1.95 \pm 0.06$ which is close to 2:1 in Fig. 1d. Thus the area ratio is quantitatively consistent with a unidirectional commensurate modulation with $\lambda = 3b$ and $\phi = 0°$ (simulations shown in Supplementary Fig. 2 visualise that other phases than $\phi = 0°$ are inconsistent with our data). Furthermore, the inferred quadrupole shift ratio (Eq. (2)) is also consistent within error bars with the expected ratio -1:2 as shown in Fig. 4c. We therefore conclude that all features of the $^{17}$O NMR line shape quantitatively support that the unidirectional modulation in the high-field CDW phase of YBa$_2$Cu$_3$O$_{6.56}$ is commensurate with a period $3b$. The real-space pattern of the charge density at O(2) sites is depicted in Fig. 2.

**Insensitivity to field and doping.** Interestingly, this result turns out to be robust from 18 to about 27 T (Fig. 4a, c), showing that the local period remains commensurate as a function of magnetic field, including across the vortex melting field $B_{melt} \simeq 23$ T at $T = 2$ K, close to an estimated value of the superconducting upper

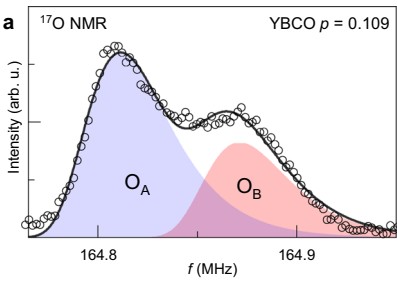

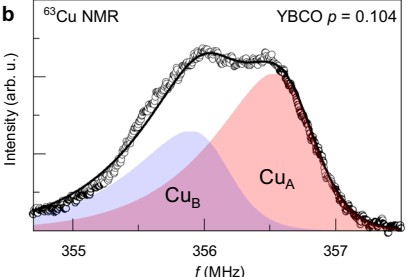

**Fig. 3 $^{17}$O and $^{63}$Cu NMR line shapes. a** Experimental second high-frequency (HF2) satellite of O(2) sites in YBa$_2$Cu$_3$O$_{6.56}$ ($p = 0.109$) in the high-field CDW phase ($T = 2$ K, $B_z = 27.1$ T). The line shape is described by two peaks O$_A$ and O$_B$ and is fitted with asymmetric line shapes of equal widths, resulting in an area ratio of $1.95 \pm 0.06$ for the two peaks. Outside the CDW phase ($T \geq 60$ K), there is only a single, symmetric line. **b** Experimental $^{63}$Cu high-frequency satellite from ref. [28] at $T = 1.3$ K and $B_z = 28.5$ T, fit with a pair of asymmetric lines (sites Cu$_A$ and Cu$_B$) of unequal areas. Fits of the whole spectrum (i.e. including all quadrupole satellites), yields a quadrupole contribution to the line splitting at O(2) sites: $^{17}\Delta\nu_{quad}/^{17}\nu_{quad} = 21/362.9$ (values in kHz) $\approx 0.058$. For Cu(2F), $^{63}\Delta\nu_{quad}/^{63}\nu_{quad} = 0.32/30.52$ (in MHz) $\approx 0.01$[28], where $\nu_{quad}$ is the effective quadrupole frequency (see Methods). Note that the asymmetry of the individual lines has a different origin for the two nuclei: a distribution of Knight shifts for $^{17}$O[41] and a distribution of $\nu_{quad}$ for $^{63}$Cu.

critical field $B_{c2} \simeq 24$ T at this doping level[45–47]. Notice that long-range, two-dimensional (2D) CDW correlations onset around $B_{CDW} \simeq 10$ T in this sample[29], becoming 3D at 17 T[30,33]. It would thus be extremely interesting to perform similar fits of the line shapes between 10 and 18 T but the much reduced peak splitting below 19 T makes the analysis unreliable.

Finally, the commensurate periodicity is not accidental for this particular hole concentration of $p = 0.109$. Consistent results from other samples (Fig. 4b, d) show that the same period-3 CDW exists from, at least, $p = 0.088$–$0.135$ (which includes O-II, O-VIII and O-III phases).

**Testing the PDW hypothesis.** In the field range over which CDW order coexists with superconductivity, we can look for indications of a putative pair-density-wave (PDW) in our data. A PDW is a peculiar superconducting state in which Cooper pairs carry a finite momentum and the superconducting order parameter varies periodically in space around a zero mean value[48]. While different PDW theories have been proposed for the cuprates[48], a robust prediction is that when PDW order coexists with uniform $d$-wave superconductivity, two CDW orders should be induced, one at $q_{PDW}$, the wave vector that characterises the superconducting modulation, and another at $2q_{PDW}$[48]. Evidence for these two modulations has recently been found in the vortex cores of Bi$_2$Sr$_2$CaCu$_2$O$_{8+\delta}$[49]. If universal, a similar phenomenon should be observed in YBCO as well, especially in the field range 15–24 T over which long-range 3D CDW and superconducting orders coexist[45].

That our low-temperature NMR spectra at $B_z$ of 18 and 21 T are fully consistent with a simple period-3 CDW (Fig. 4a, c) is evidently not favourable to the presence of a second CDW with $6b$ period. The addition of a second CDW inevitably increases the number of inequivalent sites and thus the number of peaks in the NMR spectrum. Our simulations, an example of which is shown in Fig. 5, show that, at a ~30% admixture of a $6b$ modulation,

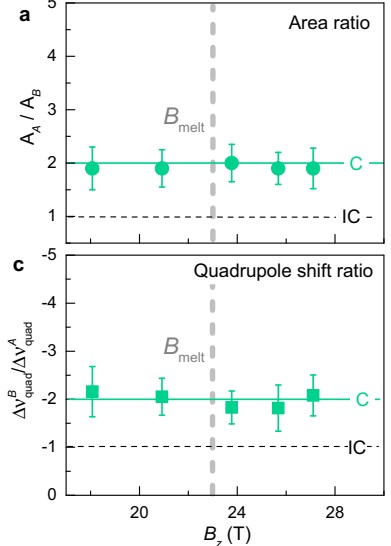

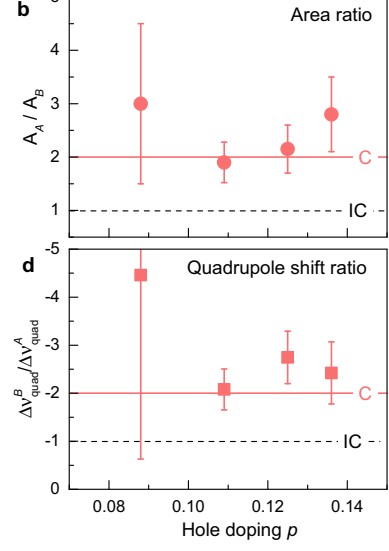

**Fig. 4 Quantitative support to the period-3 commensurate case. a, c** Field dependence of the area ratio and quadrupole shift ratio of the two peaks of the O(2) site obtained from fitting full spectra (4 quadrupole satellites), for the hole doping $p = 0.109$. **b, d** Doping dependence of the area ratio and quadrupole shift ratio at high fields (exact field values depend on sample, see Supplementary Fig. 6). Large error bars of the sample with $p = 0.088$ result from broad O(2) satellites due to necessarily imperfect O-II chain order at this doping. The continuous lines correspond to the expectation for a commensurate modulation (C) of period 3, the horizontal dashed lines correspond to the expectation for an incommensurate modulation (IC). These dashes also correspond to the expectation for a commensurate period 4, provided the phase is such that the line shape shows only two peaks, as in Supplementary Fig. 2 (panels e, j, o). Thick vertical dashed lines in (**a**) and (**c**) mark the vortex melting field at 2 K, $B_{melt} \sim 23$ T, which is in principle close to $B_{c2} = B_{melt}(T = 0$ K).

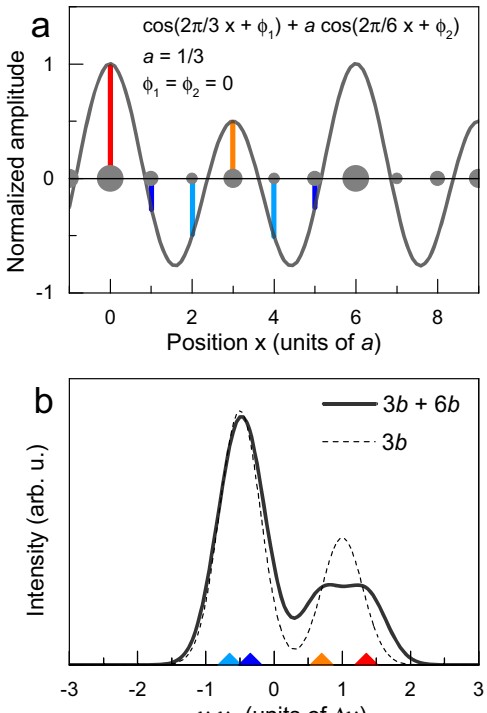

**Fig. 5 Testing the PDW hypothesis. a** Sum of uniaxial modulations with periods $3b$ and $6b$ and equal phases $\phi_1 = \phi_2 = 0$. The $6b$ modulation has amplitude $a = 1/3$. The maximal amplitude is $4/3$ but the scale has been normalised by this value. **b** Corresponding histogram for the modulation in panel (**a**). The $3b + 6b$ modulation (continuous line) shifts each of the two contributions in the $3b$ histogram (dashed line) unequally.

it becomes difficult to describe the spectrum with two peaks only. Smaller levels of admixture, on the other hand, have a sufficiently weak effect on the spectra to remain compatible with experimental data. Notice that, for the very same reasons for which any even period is inconsistent with the data (see above), the presence of two modulations with periods $4b$ and $8b$ is also ruled out.

We thus conclude that there is no conclusive evidence of a PDW state in our data. Nonetheless, there is a huge phase space that we have not explored regarding the relative amplitude and the relative phase of the two predicted charge modulations, let alone more sophisticated scenarios with complex real space patterns. To the best of our knowledge, NMR spectra have been computed in the framework of models of charge ordering[44,50] but not for realistic PDW theories. When these become available, our data could be used to further test the possibility of a PDW state in YBCO in high fields.

**Discommensurations**. In some circumstances, an incommensurate CDW may lower its energy by introducing discommensurations (DCs) which are domain walls wherein the phase $\phi$ of the order parameter (Eq. (1)) changes rapidly, while the phase in each domain is locked to a constant value[51]. Several dichalcogenides offer classical examples of a locally commensurate CDW with discommensurations, see for instance recent refs. [52–54] for NbSe$_2$.

We now discuss how such repeated phase slips in the modulation can explain why X-rays find an incommensurate wave vector in high fields while NMR locally sees a commensurate period $\lambda = 3b$. Interestingly, the presence of DCs in the high-field phase of YBCO has already been discussed in two instances: they have been proposed to have important effects on the

transport properties in high fields[11] and to be at the origin of quasiparticle scattering that leads to a skewed spatial distribution of local density of states detected in NMR[41]. Discommensurations have also been discussed in the context of holographic theories of charge order in the cuprates[55].

Following Mesaros et al.[23], the unidirectional modulation is written as

$$\Psi(x) = A \exp[i\Phi(x)] = A \exp[i(q_0 x + \varphi)] \qquad (3)$$

where $\Phi(x)$ is the phase argument that increases with the slope $q_0$. However, if the phase $\varphi$ is not constant but increases (or decreases) at repeated phase slips, then the phase argument changes to

$$\Phi(x) = q_0 x + \varphi(x) = \overline{q} x + \widetilde{\varphi}(x). \qquad (4)$$

$\overline{q}$ is the average wave vector including the effect of repeated phase slips. $\widetilde{\varphi}(x)$ contains the residual phase fluctuations which remain after $\overline{q} x$ is subtracted from the phase argument $\Phi(x)$. They will average to zero ($\overline{\widetilde{\varphi}(x)} = 0$), but can lead to additional structure in the Fourier transform. X-ray scattering would find a peak at $\overline{q}$ while the NMR spectrum, by probing the modulation locally, would be determined by $q_0$, provided that phase slips have a small density and do not extend over many lattice sites. From Eq. (4) it follows that over a length $L$ the phase argument increases due to phase slips by

$$\varphi(L) = \overline{q} L - q_0 L = \delta \cdot L, \qquad (5)$$

where the incommensurability $\delta$ is a measure of the density of phase slips: $\delta = \frac{\varphi(L)}{L}$.

For the O-II sample studied by NMR in high fields ($p = 0.109$), we estimate $q = \overline{q} = 0.323 \pm 0.005$ from zero-field X-ray studies of YBCO samples with similar hole doping[56–58] (X-ray studies did not observe a change of the wave vector as a function of field[32,33]).

The incommensurability $\delta = \overline{q} - q_0 = 0.323 - 0.333 \approx -0.01 \pm 0.005$ implies that after $L = 100b$ the acquired phase due to phase slips is only $\delta \cdot L = -0.01 \frac{2\pi}{b} \cdot 100b = -2\pi$. For $q_0 = 1/3$ possible phase slips are $\pm 2\pi/3$ (see Fig. 6a). This means that at most three phase slips per $100b$ are necessary to make a modulation locally commensurate with the lattice, if all phase slips are negative. Such a small number of discommensurations is difficult to resolve in X-ray measurements, because the intensity of satellite peaks is expected to be small. Notice that, for the sake of simplicity, we assume here a regular array of atomically sharp DCs but, in a real material, the phase slips have a finite width and, being subject to pinning by disorder, they may be ordered only on average.

Discommensurations themselves are also not expected to be visible in the NMR spectra of YBCO: If each discommensuration is small and affects the phase of the modulation in the vicinity of only one or two atoms, then no more than 5% of the NMR spectral intensity should be directly affected. Figure 6 displays the effect of negative phase slips which come in two flavours, either stretching the minimum (Fig. 6a) or the maximum (not shown) of the modulation. In the former case, the larger peak (A) increases in intensity while in the latter case the less intense peak (B) increases further.

It is also possible, in principle, to reach $\overline{q} \approx 0.32$ by adding many phase slips to a $q_0 = 1/4$ modulation. For example, starting from a CDW with $q_0 = 1/4$ and phase $\phi = 45°$ whose spectrum has two peaks of equal amplitude (Supplementary Fig. 2o), we introduce repeated phase slips of $+\pi/2$ after every third atom, which shifts the wave vector to $\overline{q} = 1/3$ (Fig. 6c). Incommensurate global wave vectors like $\overline{q} = 0.323$ are equally possible by skipping a few of the phase slips. Our calculation shows that this

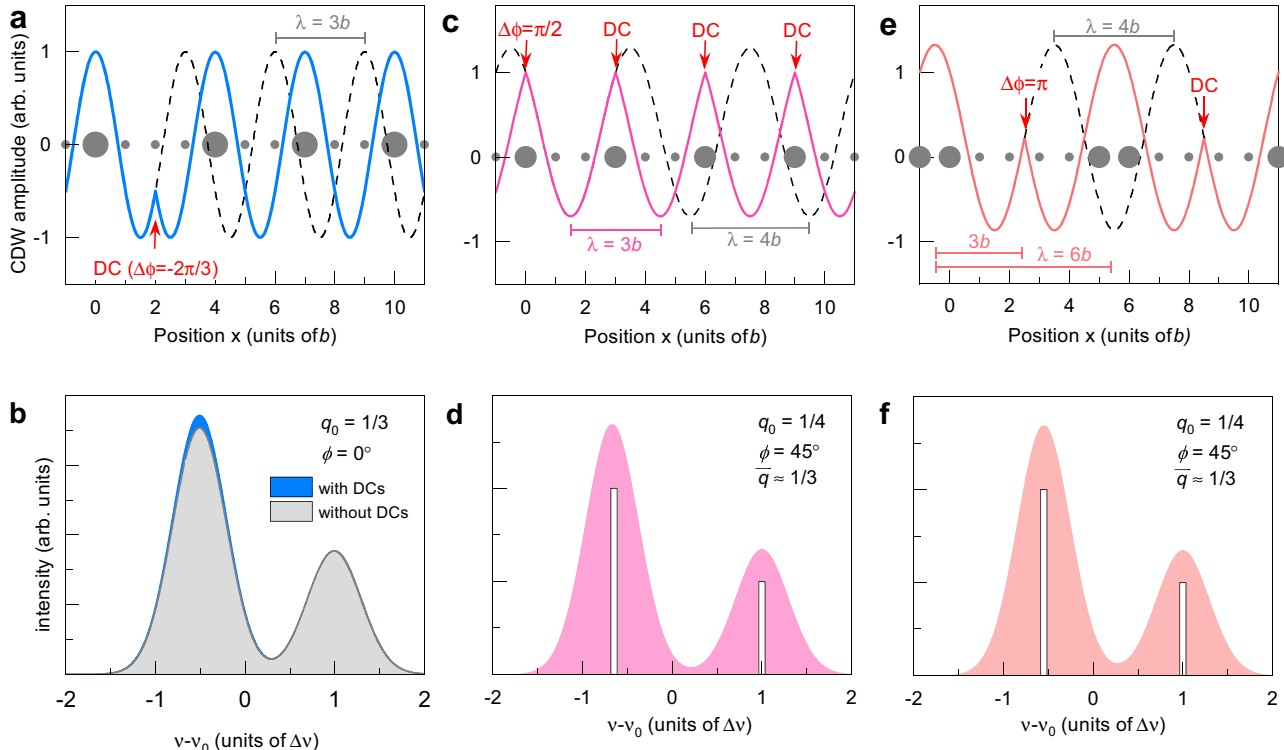

**Fig. 6 Effect of discommensurations (DC). a** A negative phase slip of $-2\pi/3$ at position 2 of an undistorted $\lambda = 3b$ modulation (black dotted line) increases the region of minimal charge density. As the effective separation between maxima increases, negative phase slips, if occurring repeatedly, lead to a longer period. **b** Histogram of the modulation shown in (**a**): $-2\pi/3$ discommensurations increase the left peak intensity but, in the diluted limit (fewer than 1 DC per 10 sites), the change in area ratio of the two peaks is too small to be unambiguously detected. **c** Starting with an undistorted $\lambda = 4b$ modulation of phase $\phi = 45°$ (black dotted line), positives phase slip of $\pi/2$ at the positions 0 and 3, etc. lead to a compressed modulation of non-sinusoidal character (continuous line) having a global wave vector $\overline{q} \approx 1/3$, i.e. period $3b$. Smoother discommensurations extending over a finite length would restore a more sinusoidal modulation. **d** Histogram of the modulation shown in (**c**). The area ratio of the two peaks is close to 2:1 but the shift ratio is $-1.54{:}1$, while the experimental data are more consistent with a shift ratio of $-2{:}1$. **e** Same idea as in (**c**) with larger but less frequent $\pi$ phase slips to shift from $q_0 = 1/4$ to the global wave vector $\overline{q} \approx 1/3$. The separation between maxima is $3b$ as in (**c**); however, maxima are inequivalent, so a superstructure with period $6b$ appears. **f** histogram of the modulation shown in (**e**). The area and shift ratios of the two peaks are both close to 2:1. Note that, in order to ensure charge neutrality, the modulation has been shifted vertically in (**c**) and (**e**) such that the integral over a period $3b$ is zero.

yields almost the same NMR spectrum as a period-3 sinusoidal modulation (compare Figs. 6d and 1d). However, the quadrupole shift ratio of the two peaks is $-1.54{:}1$ in that case, which agrees less with the data than the commensurate period-3 model giving $-2{:}1$ (Fig. 4d). Moreover, because it has one phase slip per period, the resulting modulation is probably indistinguishable from a "native" $q_0 = 1/3$ CDW for scattering techniques: in a real material, the discontinuities shown in Fig. 6c would be smoothed and the modulation probably close to sinusoidal.

The number of discommensurations can be halved if instead $+\pi$ phase slips are assumed, as shown in Fig. 6e. Although the expected NMR spectrum (Fig. 6f) is also consistent with the data (area ratio close to 2:1 and quadrupole shift ratio closer to $-2{:}1$), $+\pi$ phase slips lead to a superstructure with large and small maxima that have a period of $6b$, so X-ray scattering should find a peak at the corresponding wave vector, which has not been reported, to our knowledge.

**Consistency with Cu NMR**. The $^{63}$Cu NMR experiments that discovered the high-field CDW phase of YBCO[28] were interpreted in terms of a unidirectional CDW, propagating along the $a$ axis with a commensurate $4a$ period[28]. In fact, this proposal was aimed at providing a single explanation for two separate observations: a line splitting for Cu(2F), those planar Cu sites below oxygen-full chains (see Fig. 3), and the absence (or the much

lower magnitude) of this splitting for Cu(2E), below empty chains. However, the line splitting itself is not inconsistent with a propagation along $b$, as mentioned in ref. [28] and further exemplified by the present work.

In this case, the contrasting NMR response below empty and full chains cannot be explained by the modulation propagating along $b$ and another explanation must thus be found. This Cu(2F)/Cu(2E) dichotomy probably signifies that the presence (absence) of oxygen in the nearest chain enhances (reduces) the CDW amplitude. This is conceivable as the high-field CDW order is 3D so that chain-oxygen phonons might play a role. The Cu(2F)/Cu(2E) dichotomy implies a charge differentiation that has the same $2a$ periodicity as the O-II structure and is thus difficult to detect in X-ray scattering. We notice that a recent X-ray study found that microscopic details of CDW order in YBCO are more complex than previously thought[59], which might have a connection with this complex phenomenology.

In the next two sections, we discuss how to understand the $^{63}$Cu results quantitatively.

**Predominant oxygen character of the CDW**. Before discussing details of the $^{63}$Cu line shape, it is instructive to compare the magnitude of the charge modulation at Cu and O sites. For this, we compare the values of the relative quadrupole splitting

$\Delta\nu_{quad}/\nu_{quad}$ from the spectra shown in Fig. 3. We find that $\Delta\nu_{quad}/\nu_{quad}$ is about 6 times larger at O sites. $^{17}$O NMR is thus much more sensitive to the CDW than $^{63}$Cu NMR. Under the assumption that these numbers are proportional to the change in charge density, this means that the CDW has six times larger amplitude at O sites. If, however, this assumption is not strictly valid (see section 'CDW characteristics encoded in NMR spectra'), for instance, if the change in $\nu_{quad}$ at Cu sites mainly reflects the variation of charge density in the four bridging O$_{2p}$ orbitals, then the factor 6 should be taken as a lower bound. That the CDW modulates mainly the hole density at oxygen sites is consistent with STM measurements[60].

**Out-of-phase modulations.** We now discuss the $^{63}$Cu(2F) line shape. O(2) and Cu(2F) sites are separated by half a unit cell in the $a$ direction but since they have the same projection along the direction of propagation of the CDW ($b$ axis), one would expect their NMR line shapes to be identical. Experimentally, however, the shape of the Cu(2F) line (Fig. 3b) appears to be "inverted" with respect to that of O(2) (Fig. 3a): for Cu, it is the most intense peak which is at high frequency, with a positive quadrupole shift, while the less intense peak has a negative quadrupole shift. The area ratio 1:1.5 differs slightly from 2:1 found for $^{17}$O but this is likely due to the much larger uncertainty of $^{63}$Cu results for which signal intensities are less reliable (see "Methods") and the line splitting is barely resolved (see above). A ratio 1:2 remains consistent with the data, and the quadrupole shift ratio is $-2$:1 as for $^{17}$O. With this semi-quantitative agreement, it is clear that the $^{63}$Cu(2F) line shape is just the mirror version of the O(2) line shape. We expect this inversion to reflect a difference of charge density at Cu sites, not a difference in the way Cu and O nuclei couple to the CDW. Indeed, the Knight shift and the quadrupole frequency of $^{63}$Cu both increase with increasing hole density, exactly as they do for $^{17}$O.

As shown in Fig. 1e, f, there is a simple way to explain the mirrored $^{63}$Cu line shape: the same commensurate period-3 CDW only needs to be inverted, that is, it is dephased by $\delta\phi = \pi$. This phase shift amounts to a translation by $3b/2$ with respect to the modulation at O(2) and it is easy to see from Fig. 1e,f that a shift by any odd multiple of $b/2$ actually yields the same histogram. In other words, we find that the CDW at Cu sites has a phase difference $\delta\phi = (2n + 1)\pi/3 = \pi/3, \pi$ or $5\pi/3$ with respect to the CDW at those O sites which are perpendicular to the propagation vector.

Of particular interest is the value $\delta\phi = \pi$ because there has been much discussion as to whether a similar anti-phase relationship exists between O sites in perpendicular bonds (*i.e.* O(2) and O(3) here). Such a $d$-symmetry form factor of the CDW at oxygen sites has been predicted by a number of theories (for example[19–21,61]) and identified in STM studies of Bi$_2$Sr$_2$CaCu$_2$O$_{8+x}$ and Ca$_{2-x}$Na$_x$CuO$_2$Cl$_2$[62] but it has been controversial in YBCO from X-ray studies in zero-field[59,63,64].

Here, the possibility that Cu and O(2) are in anti-phase has a twofold consequence: either Cu and O(3) are also in anti-phase, in which case O(2) and O(3) must be in-phase and thus there is no $d$ form factor. Or O(2) and O(3) are in anti-phase and thus O(3) is in-phase with Cu. We cannot confidently decide between these two possibilities from our data alone because the overlap of O(3F) and O(3E) lines hampers a quantitative analysis of the O(3) line shapes.

As shown in Supplementary Fig. 4a, b and discussed in Supplementary Note 4, the O(3) satellite spectrum can be fit assuming either in-phase or anti-phase relationships between O(2) and O(3) and furthermore there is significant uncertainty on the value of the splitting at O(3E) and O(3F) sites. Nevertheless,

a $d$ form factor may be considered to be more likely because (1) $\delta\phi = \pi$ is highly probable ($\delta\phi = \pi/3$ and $5\pi/3$ have never been observed in cuprates, either experimentally or theoretically), (2) it makes more physical sense that the charge density is modulated in-phase in those orbitals that are strongly hybridised along the direction of propagation, namely O(3) is in-phase with Cu. The phase at O(2) sites would then follow from the leading tendency to have out-of-phase modulations between nearest neighbour oxygen sites, (3) the error of the fitting of O(3) satellites is smallest for anti-phase ($d$ symmetry) intra-unit-cell order, see Supplementary Fig. 4c (keeping in mind that the difference is small and that there is uncertainty on how to fit these spectra, as explained in Supplementary Note 4). Under this assumption of a $d$ form factor, we propose in Fig. 2b, c a real-space pattern of the charge density consistent with our NMR data.

We notice that in theories of a unidirectional CDW with a $d$ form factor, which O is in-phase or in anti-phase with Cu appears to depend on the details of the model[19,65]. Our observation of an out-of-phase relationship between O(2) and Cu is thus an important piece of information to understand the microscopic origin of the CDW. We also suggest that it would be interesting to introduce a small planar anisotropy in the parameters of theoretical models in order to investigate whether orthorhombicity plays a role in determining the relative phase between different lattice sites.

## Discussion

Let us first summarise our main result at the qualitative level: the 3D long-range CDW in YBCO is unidirectional (consistent with previous studies[28,32–35]) and, at the local scale probed by NMR, commensurate with the lattice. The incommensurability found in X-ray scattering studies must thus be explained by the presence of an array of discommensurations. We conjecture that the 2D short-range CDW also has a commensurate local period and discommensurations, even though this cannot be tested directly by NMR because the short-range order only leads to featureless line broadening.

We now discuss the consequences of this finding.

The presence of discommensurations in an otherwise commensurate modulation is in general the result of a compromise between two opposite, commensurate and incommensurate, tendencies. Here in YBCO, as well as in other cuprates, the Fermi surface has been suspected to dictate the overall global periodicity ($\overline{q}$), that is in general found to be incommensurate and decreasing with doping. This includes the possibility that the wave vector is determined by the $q$-space structure of the electron–phonon coupling (that itself depends on the Fermi surface)[66]. Therefore, our finding does not necessarily questions previous claims that the Fermi surface plays a role in determining the global CDW periodicity. It must be noted, however, that incommensuration alone does not constitute definitive evidence that the CDW wave vector is determined by the Fermi surface. In particular, long-range Coulomb interaction may favour the formation of a Wigner crystal, which is another form of charge order that is in general incommensurate. In the absence of short-range correlation effects, the ordering wave vector for a Wigner crystal should increase with increasing doping, at variance with experiments. However, the interplay between correlation effects at short distances and Coulomb repulsion at long distances may well result in an incommensurate CDW with nontrivial doping dependence (see for instance refs. [67–69]). In any event, it is important to realise that we are not opposing a locally commensurate modulation that has discommensurations and a "genuinely" incommensurate wave that has a perfectly sinusoidal modulation. They represent two different ways of bringing about the same incommensurate

wave vector, two limiting cases of the same model in which one varies the width of the DCs.

The question then is: where does commensuration originate from? The commensurate local periodicity $(2\pi/q_0)$ can in principle result either from a lock-in to the lattice or from an intrinsically commensurate ordering mechanism. Three scenarios can be considered:

• Scenario 1: the modulation simply adjusts itself locally to the lattice (lock-in) with a commensurate period whilst adding phase slips to maintain the dictated global periodicity. The local period is chosen according to the Fermi surface in such a way that the necessary number of discommensurations is minimised, i.e. $q_0$ locks to the commensurate fraction $m/n$ the closest to $\bar{q}$. For YBCO in the range $p \simeq 0.08-0.16$, $m/n = 1/3$ and at a doping $p = 0.109$, $-2\pi/3$ phase slips are separated by an average distance of thirty unit cells. Mesaros et al. argue against such a lattice locking because $q_0 = 1/4$ is observed in Bi2212 at doping levels ranging from $p = 0.06$ to $0.17$, without any jump to $q_0 = 1/3$ or $1/5$[23]. However, whether lock-in to $1/3$ or $1/5$ should actually be observed is unclear as there does not appear to be definitive evidence that global $\bar{q}$ values for $p = 0.06$ ($p = 0.17$) are closer to $1/3$ ($1/5$) than to $1/4$, within experimental error bars[2]. Notice that this scenario says nothing on the possibility that charge ordering has a fundamental local period four. It only says that, in YBCO, the CDW reduces its energy more by locking to the lattice.

• Scenario 2: $q_0$ locks to $1/4$ in YBCO, the fundamental wave vector invoked for Bi-based cuprates[23–25], but in order to reach $\bar{q} \simeq 0.32$, discommensurations are so frequent that the charge modulation is effectively best described as a period-3 CDW for $\pi/2$ DCs or a CDW with a period-6 motif for $\pi$ DCs, as shown in Fig. 6c, e. Both can be seen as variants of a three-unit-cell motif that produces only two inequivalent sites, as required by the NMR spectra. As mentioned above, the main problem with this scenario is that, for $\pi/2$ DCs, the NMR splitting is not perfectly captured and, for $\pi$ DCs, the expected satellite peaks in X-ray scattering[23] have not been observed.

• Scenario 3 cannot be distinguished from scenario 1 in our experiments as it differs only in the physical origin of the local commensuration: a commensurate ordering mechanism, driven by strong correlations, is at work but, for some reason, it leads to $q_0 = 1/3$ in YBCO, not $1/4$ as in Bi-based cuprates. One might object that this scenario would be at odds with the rare presence of solutions with $q_0 = 1/3$ in numerical studies of the Hubbard model at doping levels $p \sim 0.1$ and in the strong repulsion limit (refs. [70–73] and references therein). However, there are multiple factors, not necessarily included in these calculations, that play a role in the final $q_0$ value: the electron–phonon coupling, the long-range Coulomb interactions, the (ordered or fluctuating) nature of spin correlations, etc. Predicting the CDW wave vector is a notoriously difficult problem[74]. In Bi-based cuprates, the strongest argument in favour of $q_0 = 1/4$ being a fundamental ordering wave vector and not a lock-in effect probably lies in the STM observation of a purely commensurate $q = 1/4$ (no DCs) at very low doping in the insulating/magnetic phase of Bi2201[26,27]. However, given the significant differences in crystal structures and hole doping levels, there is too much of a gap between insulating Bi2201 and metallic YBCO to make a safe extrapolation, let alone the possibility that the surface (probed in Bi2201) and the bulk (probed here in YBCO) behave somewhat differently[75].

Summarising this discussion, the simple splitting of NMR lines in YBCO reveals a bimodal charge distribution that is best explained by a three-unit-cell elementary motif containing two inequivalent sites. While we cannot fully exclude that this motif emerges from a fundamental four-unit-cell local period (scenario 2), the scenario that appears to be both the simplest and the most consistent with our NMR results is a commensurate local periodicity of three unit cells with a small number of discommensurations.

Apart from these quantitative aspects, we interpret our results as evidence that there is no fundamental difference between YBCO and Bi-based cuprates: both show a locally commensurate CDW but an incommensurate global wave vector. The wave primarily modulates the oxygen hole density, with the O sites in orthogonal directions being in anti-phase in the case of Bi cuprates and probably for YBCO as well. There is also mounting evidence that the CDW in zero field is bi-directional at long length scales but unidirectional at short length scales (comparable to the CDW period) in both compounds[16–18,60,62]. What mostly differentiates YBCO from the Bi compounds is the much lower level of disorder and the stronger orthorhombicity. This results in three times longer correlation lengths at $T \simeq T_c$, stronger competition with superconductivity[46,47] and thus, in turn, the unique possibility to observe purely unidirectional, long-range order in high magnetic fields[29]. We note that field-induced CDW order is also observed in $Bi_2Sr_{2-x}La_xCuO_6$[9] but its relationship to the high-field CDW of YBCO is at present unclear.

It is interesting to note that the stripe order of La-based cuprates, while often considered as a distinct type of CDW, is in fact not fundamentally different even at a simple phenomenological level. The CDW is unidirectional, though forming orthogonal domains in the absence of low-temperature structural distortion[76]. $\bar{q}_{CDW}$ is incommensurate, even for $p = 0.125 = 1/8$ doping where commensurability effects have been thought to be most relevant. Also, it has been argued that discommensurations account for the incommensurability at $1/8$[23,77]. Furthermore, recent work[78–80] reveals that the increase of $\bar{q}_{CDW}$ with doping[81–83], that is frequently considered a major difference with other cuprates, is actually a consequence of the mutual locking of long-range spin and charge orders at low temperatures, a situation that never occurs in other cuprates (see also ref. [84] for a theoretical perspective on these questions).

We take these considerations as suggestive evidence that superconducting cuprates show, over a substantial doping range, a generic instability towards the formation of a unidirectional and locally commensurate CDW whose global periodicity is nonetheless incommensurate. The high-field phase of YBCO studied here offers a unique realisation of this CDW phase, whose observation is in general complicated by the conspiracy of disorder, magnetic ordering and competition with superconductivity.

## Methods

**Samples.** High quality $^{17}O$-enriched detwinned YBCO single crystal samples were grown from self-flux[85]. The same crystals were used and characterised in our previous NMR studies[17,29,41,43,45,46,86]. Sample information is summarised in Supplementary Table 1.

**NMR.** Frequency swept NMR spectra were acquired by summing Fourier-transformed echoes measured with standard spin-echo sequences on home-built heterodyne spectrometers at LNCMI Grenoble. Fields above 20 T were provided by resistive magnets at LNCMI.

$^{17}O$ has several advantages with respect to $^{63}Cu$ NMR. First, $^{17}O$ spectra in the high-field CDW state are better resolved and have larger signal-to-noise ratio[29,41]. Second, it is experimentally much easier to record $^{17}O$ spectra free of distortion by $T_2$ effects. This is because the relaxation time $T_2$ of $^{17}O$ nuclei is much longer than for $^{63}Cu$, owing to both a smaller value of the hyperfine field and a symmetric position that filters out the contribution of antiferromagnetic fluctuations to the longitudinal and transverse relaxation rates $T_1$ and $T_2$.

For the O-II chain-oxygen superstructure there are two types of O(3) planar sites, namely O(3E) and O(3F) below oxygen-empty and oxygen-filled chains, respectively, but only one O(2) site. To separate O(2) and O(3) planar sites while keeping a large $c$ axis component of the external field, $B_z$, most samples were tilted by $\simeq 16°$ towards the $b$ axis. For YBCO $p = 0.109$, this angle was $\simeq 18°$ and leads to a quadrupole frequency $\nu_{quad,0} = 362.9$ kHz (defined as the frequency difference between two consecutive satellite lines) in the absence of long-range CDW order.

**Analysis**. The ability to extract information on the CDW from NMR spectra depends on three factors, themselves depending on the material and the nucleus considered: (i) whether the CDW-induced changes are large enough with respect to the natural line broadening (that depends on chemical homogeneity and quenched disorder), (ii) whether the different crystallographic sites can be separated, (iii) whether the magnetic hyperfine and quadrupole effects can be disentangled (both types of interaction affect the energy levels of any nuclear spin $I > 1/2$). In our previous works, we have shown how to extract relevant information from $^{63}$Cu[28] and $^{17}$O[17,29,41] spectra in YBa$_2$Cu$_3$O$_y$: an increase (decrease) in the local charge density at site $x$ leads to an increase (decrease) of both the Knight shift $K(x)$ and the quadrupole shift $v_{quad}(x)$. Note that $v_{quad}$ is defined here as half the distance between first high and low-frequency satellite lines ($m_I = \pm 1/2$ to $\pm 3/2$ transitions) thus $v_{quad}$ differs in general from the standard quadrupole frequency $v_Q$.

The two contributions $\Delta K(x)$ and $\Delta v_{quad}(x)$ combine additively so that each nucleus experiences a change of its resonance frequency with respect to the original resonance without CDW order:

$$\Delta v_{tot}(x) = \gamma B \Delta K(x) + n \cdot \Delta v_{quad}(x), \qquad (6)$$

where $n$ is an integer between $-2$ and $+2$ depending on the particular low- or high-frequency satellite in the $^{17}$O-NMR spectrum (the central line is not used as the contributions from different crystallographic sites cannot be separated). HF2, the outer high-frequency satellite ($n = +2$), is thus most sensitive to the modulation because the resulting total splitting $\Delta v_{tot}$ is largest. LF1 and LF2 are much less affected by the CDW because the minus sign in Eq. (6) leads to a near compensation of the two terms that have comparable magnitude. Yet, we fit all four satellites (LF2, LF1, HF1 and HF2) simultaneously in order to further constrain the analysis and improve its reliability (see also ref. [41]).

## Data availability
The data that support the findings of this study are available from the corresponding authors upon reasonable request.

## Code availability
The codes that support the findings of this study are available from the corresponding authors upon reasonable request.

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

# ARTICLE

52. Soumyanarayanan, A. et al. Quantum phase transition from triangular to stripe charge order in $NbSe_2$. *Proc. Natl Acad. Sci. USA* **110**, 1623–1627 (2013).

53. Feng, Y. et al. Itinerant density wave instabilities at classical and quantum critical points. *Nat. Phys.* **11**, 865–871 (2015).

54. Pásztor, A. et al. Holographic imaging of the complex charge density wave order parameter. *Phys. Rev. Res.* **1**, 033114 (2019).

55. Andrade, T., Krikun, A., Schalm, K. & Zaanen, A. Doping the holographic Mott insulator. *Nat. Phys.* **14**, 1049–1055 (2018).

56. Hücker, M. et al. Competing charge, spin, and superconducting orders in underdoped $YBa_2Cu_3O_y$. *Phys. Rev. B* **90**, 054514 (2014).

57. Blanco-Canosa, S. et al. Momentum-dependent charge correlations in $YBa_2Cu_3O_{6+\delta}$ superconductors probed by resonant x-ray scattering: evidence for three competing phases. *Phys. Rev. Lett.* **110**, 187001 (2013).

58. Blanco-Canosa, S. et al. Momentum-dependent charge correlations in $YBa_2Cu_3O_{6+\delta}$, superconductors probed by resonant x-ray scattering: evidence for three competing phases. *Phys. Rev. B* **90**, 054513 (2014).

59. McMahon, C. et al. Orbital symmetries of charge density wave order in $YBa_2Cu_3O_{6+x}$. *Sci. Adv.* **60**, eaay0345 (2020).

60. Kohsaka, Y. et al. An intrinsic bond-centered electronic glass with unidirectional domains in underdoped cuprates. *Science* **315**, 1380–1385 (2007).

61. Fischer, M. H., Wu, S., Lawler, M., Paramekanti, A. & Kim, E.-A. Nematic and spin-charge orders driven by hole-doping a charge-transfer insulator. *New J. Phys.* **16**, 093057 (2014).

62. Fujita, K. et al. Direct phase-sensitive identification of a *d*-form factor density wave in underdoped cuprates. *Proc. Natl Acad. Sci. USA* **111**, E3026–E3032 (2014).

63. Comin, R. et al. The symmetry of charge order in cuprates. *Nat. Mater.* **14**, 796–800 (2015).

64. Forgan, E. M. et al. The microscopic structure of charge density waves in underdoped $YBa_2Cu_3O_{6.54}$ revealed by X-ray diffraction. *Nat. Commun.* **6**, 10064 (2015).

65. Thomson, A. & Sachdev, S. Charge ordering in three-band models of the cuprates. *Phys. Rev. B* **91**, 115142 (2015).

66. Banerjee, S., Atkinson, W. A. & Kampf, A. P. Emergent charge order from correlated electron-phonon physics in cuprates. *Commun. Phys.* **3**, 161 (2020).

67. Löw, U., Emery, V. J., Fabricius, K. & Kivelson, S. A. Study of an Ising model with competing long- and short-range interactions. *Phys. Rev. Lett.* **72**, 1918 (1994).

68. Schulz, H. J. Wigner crystal in one dimension. *Phys. Rev. Lett.* **71**, 1864–1867 (1993).

69. Boschini, F., Minola, M. & Sutarto, R. et al. Dynamic electron correlations with charge order wavelength along all directions in the copper oxide plane. *Nat. Commun.* **12**, 597 (2021).

70. Tu, W. L. & Lee, T. K. Genesis of charge orders in high temperature superconductors. *Sci. Rep.* **6**, 18675 (2016).

71. Zheng, B.-X. et al. Stripe order in the underdoped region of the two-dimensional Hubbard model. *Science* **358**, 1155–1160 (2017).

72. Huang, E. W., Mendl, C. B., Jiang, H., Moritz, B. & Devereaux, T. P. Stripe order from the perspective of the Hubbard model. *npj Quant. Mater.* **3**, 22 (2018).

73. Ponsioen, B., Chung, S. S. & Corboz, P. Period 4 stripe in the extended two-dimensional Hubbard model. *Phys. Rev. B* **100**, 195141 (2019).

74. Johannes, M. D. & Mazin, I. I. Fermi surface nesting and the origin of charge density waves in metals. *Phys. Rev. B* **77**, 165135 (2008).

75. Brown, S., Fradkin, E. & Kivelson, S. Surface pinning of fluctuating charge order: an extraordinary surface phase transition. *Phys. Rev. B* **71**, 224512 (2005).

76. Choi, J. et al. Disentangling intertwined quantum states in a prototypical cuprate superconductor, https://arxiv.org/abs/2009.06967 (2020)

77. Tranquada, J. M., Ichikawa, N. & Uchida, S. Glassy nature of stripe ordering in $La_{1.6-x}Nd_{0.4}Sr_xCuO_4$. *Phys. Rev. B* **59**, 14712 (1999).

78. Miao, H. et al. High-temperature charge density wave correlations in $La_{1.875}Ba_{0.125}CuO_4$ without spin-charge locking. *Proc. Natl. Acad. Sci. USA* **114**, 12430–12435 (2017).

79. Miao, H. et al. Incommensurate phonon anomaly and the nature of charge density waves in cuprates. *Phys. Rev. X* **8**, 011008 (2018).

80. Miao, H. et al. Formation of incommensurate charge density waves in cuprates. *Phys. Rev. X* **9**, 031042 (2019).

81. Yamada, K. et al. Doping dependence of the spatially modulated dynamical spin correlations and the superconducting-transition temperature in $La_{2-x}Sr_xCuO_4$. *Phys. Rev. B* **57**, 6165 (1998).

82. Hücker, M. et al. Stripe order in superconducting $La_{2-x}Ba_xCuO_4$ (0.095 ≤ x ≤ 0.155). *Phys. Rev. B* **83**, 104506 (2011).

83. Fink, J. et al. Phase diagram of charge order in $La_{1.8-x}Eu_{0.2}Sr_xCuO_4$ from resonant soft x-ray diffraction. *Phys. Rev. B* **83**, 092503 (2011).

84. Nie, L., Maharaj, A. V., Fradkin, E. & Kivelson, S. A. Vestigial nematicity from spin and/or charge order in the cuprates. *Phys. Rev. B* **96**, 085142 (2017).

85. Liang, R., Bonn, D. A. & Hardy, W. N. Growth of YBCO single crystals by the self-flux technique. *Philos. Mag.* **92**, 2563–2581 (2012).

86. Vinograd, I. et al. Nuclear magnetic resonance study of charge density waves under hydrostatic pressure in $YBa_2Cu_3O_y$. *Phys. Rev. B* **100**, 094502 (2019).

## Acknowledgements

The authors thank A. Mesaros, S.A. Kivelson, W.A. Atkinson, J. Chang, P. Monceau and S. Fratini for valuable discussions. This work was performed at the LNCMI, a member of the European Magnetic Field Laboratory. Work in Grenoble was supported by the Laboratoire d'Excellence LANEF (ANR-10-LABX-51-01) and by the French Agence Nationale de la Recherche (ANR) under reference ANR-19-CE30-0019 (Neptun).

## Author contributions

I.V., R.Z., M.H. and T.W. performed experiments. H.M., S.K. and M.H.J. assisted in experiments. I.V., R.Z. and M.H. analysed data. I.V. and M.H. performed numerical simulations. D.A.B., W.N.H. and R.L. grew single crystals. I.V. and M.H.J. wrote the paper. All authors commented on the manuscript.

## Competing interests

The authors declare no competing interests.
