## [Peer Review File · Nature Communications]

REVIEWER COMMENTS

Reviewer #1 (Remarks to the Author):

Vinograd et al report an analysis of 170 NMR spectra in $\text{YBa}_2\text{Cu}_3\text{O}_y$ (YBCO) and conclude that the charge density wave (CDW) in underdoped YBCO is commensurate with lattice, with a three-unit-cell periodicity and unidirectional.

The wave vector of the CDW has been controversial. Previous X-ray scattering measurements reported an incommensurate CDW and scanning tunnel microscopy (STM) claimed a four-unit-cell periodicity. In fact, the authors of this paper also concluded a four-unit-cell periodicity in their early paper (Wu et al, Nature 2011). In the present manuscript, the authors provide an explanation, namely, a phase slips scenario, to reconcile the X-ray data. They also re-analyzed their Cu-NMR data published in 2011 and concluded that the data can also be fitted by the three-unit-cell periodicity.

These findings are useful in understanding the CDW in cuprates and can be potentially important.

There are some inconsistency and incompleteness of the current manuscript. For example, the authors state that an incommensurate wave vector $q = 0.323 \pm 0.005$ for YBCO with $p = 0.11$ by X-ray scattering in high fields finds should give a spectrum with two peaks of equal intensities and an asymmetric shape of the individual peaks, as shown in Fig. 1b. However, the simulation shown in Fig. 1b is for $q = 0.291$. The authors should show a simulation with $q = 0.323$. Also, a comparison with other systems such as Bi2201 should be made. The authors correctly state that previous studies have suggested CDW with possible connections to the pseudogap and superconducting phases. In fact, CDW competes with superconductivity in YBCO, but seems to coexist with superconductivity in Bi2201 (Ref.[73]). The doping dependence of the wave vector also seems to be different (Peng et al, Nature Materials, 2018). These differences should be mentioned in Introduction when the topic is touched.

Reviewer #2 (Remarks to the Author):

In this paper, the authors report the presence of the commensurate period of three unit cells at the long-range CDW order in YBCO_y , estimated from the analyses of the NMR spectrum in the high field from 18 to 27 T. They also show that this scenario can also interpret the 63Cu -NMR results initially considered as the evidence of the period-4 CDW. It is considered to be a scientific progress that the physical picture is revised to the right direction after the results obtained in higher-quality samples and by more convincing techniques. The authors carefully calculated the NMR spectrum based on the various possible scenario. The discussion done in the paper is useful, and I agree that this paper is published.

The authors encourage to consider my comment:

1. The analyses of the NMR spectra seem to be consistent with the X-ray scattering by Blackburn et al. As pointed, X-ray commensurability was observed at $H=0$, but the NMR measurement was performed in high-field. How do we understand the absence of the spectrum broadening due to the CDW ordering in low field? This is just due to the sensitivity of the NMR measurements?
2. The origin of the three unit cell periodicity.

Reviewer #1:

The authors state that an incommensurate wave vector $q = 0.323 \pm 0.005$ for YBCO with $p = 0.11$ by X-ray scattering in high fields finds should give a spectrum with two peaks of equal intensities and an asymmetric shape of the individual peaks, as shown in Fig. 1b. However, the simulation shown in Fig. 1b is for $q = 0.291$. The authors should show a simulation with $q = 0.323$.

We thank the reviewers for raising this point that we indeed did not explain clearly. Any incommensurate wave vector, whatever its value, leads to the exact same NMR line shape (the two peaks of equal intensities with an asymmetric shape of the individual peaks). So, which q value we take does not matter for the NMR spectrum, provided it is incommensurate. It turns out that we chose q as small as 0.291, instead of $q = 0.323$, for Fig. 1a because this allows to see that the modulation is incommensurate (i.e. it yields a large - actually infinite - number of inequivalent sites) even if showing a section with length of only 11 unit cells. To have the same visual effect with $q = 0.323$ would require to plot the modulation over about four times the length (40 unit cells). To illustrate the point, we reproduce below the modulation with $q = 0.323$ together with the corresponding NMR spectrum. In the case of $q = 0.291$, the large number of inequivalent sites is not visually evident for the “positive” (red) sites. We believe this could lead some readers to misunderstand the point.

Furthermore, one should consider that the paper is not only about $q = 0.323$ as found for $p = 0.11$ but about a range of dopings $0.088 < p < 0.136$ and thus a range of q values.

In the submitted version of the manuscript, this independence of the NMR line shape with respect to the value of the (incommensurate) wave vector was alluded to in the following sentence in the caption to Fig. 1: “the incommensurate nature implies that the charge modulation is sampled by nuclei at an infinite number of different values in between the two extrema, regardless of the values of both the period and the phase”.

However, the remark of the referee makes clear that this sentence is insufficient. Therefore, we have now added the following sentences in order to make sure that there is no ambiguity:

- In the figure caption: *“Any incommensurate wave vector, whatever its value, leads to the exact same NMR line shape. For this figure, we thus use $q = 0.291$, a value slightly smaller than the experimental value $q = 0.323$ for $p = 0.11$, because this makes it easier to visualize the inequivalent sites in the 11-site chain segment.”*
- In the main text, after *“This should give a spectrum with two peaks of equal intensities and an asymmetric shape of the individual peaks, as shown in Fig. 1b”*, we have now added *“(notice that we use $q = 0.291$, a value slightly smaller than the experimental value $q = 0.323$, because this makes it easier to visualize the inequivalent sites in the 11-site chain segment in Fig. 1a. This has no impact on the simulated line shape in Fig. 1b since the histogram is independent of the value of q , provided it is incommensurate. The line shape only depends on the dimensionality (whether the CDW is uni- or bi-directional) and on the shape of the modulation (sinusoidal, square-like, solitonic-type, etc.)”*.

A comparison with other systems such as Bi2201 should be made. The authors correctly state that previous studies have suggested CDW with possible connections to the pseudogap and superconducting phases. In fact, CDW competes with superconductivity in YBCO, but seems to coexist with superconductivity in Bi2201 (Ref. [73]). The doping dependence of the wave vector also seems to be different (Peng et al, Nature Materials, 2018). These differences should be mentioned in Introduction when the topic is touched.

We believe that the referee’s point is interesting but not entirely accurate here.

First, the competition between CDW and superconductivity is not the subject of this paper: it is only mentioned once, very briefly, at the end of the paper (page 10) and not in the introduction.

Second, we do not see any evidence that Bi2201 and YBCO present qualitative differences in the way CDW and SC compete:

- to the best of our understanding (and as confirmed in a private communication with the authors of ref. 73), ref. 73 does not actually provide evidence against a competition (actually, the field effect rather suggests otherwise).
- the coexistence between CDW and SC is also present in YBCO over a substantial range of fields and temperatures. The two orders compete but coexist, in both YBCO and Bi2201.
- the doping dependence of the global wave vector is actually very similar in YBCO and Bi2201: it continuously decreases with increasing doping (see Fig. 8 in Ref. 1), as stated in our introduction: *“According to scattering experiments, Y-, Bi- and Hg-based cuprates show short-range, bi-directional (...) CDW order with an incommensurate wave vector q that decreases upon increasing hole doping p .”*
- In the paper of Peng et al. cited by the referee, the CDW order is observed in the strongly overdoped regime, a region of the phase diagram that has not be

explored in YBa₂Cu₃O_y because it cannot be reached by oxygen doping. Therefore, no direct comparison can be made here. In any event, the value of the wave vector measured by Peng et al. is in complete agreement with this decreasing trend and is found to fall on a linear extrapolation of the data at low doping.

To the best of our knowledge, there is thus no evidence that the CDW does not compete with superconductivity in Bi₂201. Certainly, the competition is not as clear as in YBCO where it is more spectacular than in any other cuprate. This is most likely because the CDW is stronger (it has a longer correlation length, as mentioned in page 10) than in Bi₂201, Bi₂212 or Hg₁201.

For all these reasons, and even though this is obviously an important issue, we are afraid that adding a discussion of the competition aspect would lengthen the paper without bringing conclusive insight and it would ultimately be confusing for a number of readers.

Reviewer #2 (Remarks to the Author):

1. How do we understand the absence of the spectrum broadening due to the CDW ordering in low field? This is just due to the sensitivity of the NMR measurements?

We thank the referee for giving us the opportunity to clarify this point. We do observe a line broadening due to the short-range CDW, in the normal state and/or in low fields. This is fully described in our ref. 15, Wu et al. Nat. Commun. 6, 6438 (2015). So, there is no sensitivity problem of NMR. However, we cannot extract much quantitative information from this featureless broadening. This is why we analyze the long-range ordered phase in high fields which has a stronger effect, leading to split NMR lines.

Please notice that the last sentence of page 4 mentions that the short-range CDW only broadens the NMR spectra symmetrically and so does not modify the overall line shape in the long-range ordered phase.

2. The origin of the three unit cell periodicity.

Since the commensurate aspect is discussed at length (page 10) in the paper, the referee is probably asking why the global period is around three and not around 4 or 5 for instance. We wrote in the beginning of page 10: *“Here in YBCO, as well as in other cuprates, the Fermi surface is likely to dictate the overall global periodicity (q), that is in general found to be incommensurate and decreasing with doping. (...) Note, however, that incommensurability may not necessarily be related to the Fermi-surface, see for instance refs. [65, 66].”*

We acknowledge that this is not quite clear. Therefore, in order to be more explicit about what we have in mind (the global period is set either by the Fermi surface or by more complex effects such as involving long-range Coulomb interactions), we rewrote this part as follows: *“Here in YBCO, as well as in other cuprates, the Fermi surface has been suspected to dictate the overall global wave vector (q), that is in general found to be*

incommensurate and decreasing with doping. (...) It must be noted, however, that incommensuration alone does not constitute definitive evidence that the CDW wave vector is determined by the Fermi surface. In particular, long-range Coulomb interaction may favor the formation of a Wigner crystal, which is another form of charge order that is in general incommensurate. In the absence of short-range correlation effects, the ordering wave vector for a Wigner crystal should increase with increasing doping. However, the interplay between correlation effects at short distances and Coulomb repulsion at long distances may well result in an incommensurate CDW with nontrivial doping dependence (see for instance refs. [65, 66, 67])."

The new ref. 67 is an excellent and very relevant paper that after submission of our manuscript:

Boschini, F., Minola, M., Sutarto, R. *et al.* Dynamic electron correlations with charge order wavelength along all directions in the copper oxide plane. *Nat Commun* **12**, 597 (2021).

We believe these modifications make clear that what exactly determines the CDW periodicity is still an open question.

REVIEWERS' COMMENTS

Reviewer #1 (Remarks to the Author):

The authors' reply to my first point is satisfactory. I suggest that the figure presented in the rebuttal be shown as Supplement from which readers will benefit.

As for the second point, the authors admit that a comparison between YBCO and other systems "is obviously an important issue". In fact, they state in their reply that the doping dependence of the global wave vector is actually very similar in YBCO and Bi2201. I thought adding some discussion would be useful, but the reasoning of paper length is acceptable under the condition that the authors add two references after Ref. [2-7] in Line 6 of the beginning of the paper; re-numbering Ref. [74] of the revised manuscript as Ref. [8], and citing Peng et al's paper as Ref. [9]. From a revised sentence such as "with possible connections to the pseudogap and superconducting phases [2-9]", readers will be self-guided to explore the issue with this minimum revision.

Reviewer #2 (Remarks to the Author):

The author responded to my comments appropriately. I recommend the manuscript is published.